# Succession of the intestinal bacterial community in Pacific bluefin tuna (*Thunnus orientalis*) larvae

Akito Taniguchi[1☯], Ryuichiro Aoki[2☯], Isamu Inoue[3], Mitsuru Eguchi[1,4]*

**1** Faculty of Agriculture, Kindai University, Nara, Japan, **2** Graduate School of Agriculture, Kindai University, Nara, Japan, **3** Aquaculture Technology and Production Center, Kindai University, Susami, Wakayama, Japan, **4** Agricultural Technology and Innovation Research Institute, Kindai University, Nara, Japan

☯ These authors contributed equally to this work.
* eguchi@nara.kindai.ac.jp

**Data Availability Statement:** All relevant data are within the manuscript and its Supporting Information files.

**Funding:** ME, the Global COE program "International education and research center for

## Abstract

We investigated the succession process of intestinal bacteria during seed production in full-cycle aquaculture of Pacific bluefin tuna (*Thunnus orientalis*). During the seed production, eggs, healthy fish, rearing water, and feeds from three experimental rounds in 2012 and 2013 were collected before transferring to offshore net cages and subjected to a fragment analysis of the bacterial community structure. We identified a clear succession of intestinal bacteria in bluefin tuna during seed production. While community structures of intestinal bacteria in the early stage of larvae were relatively similar to those of rearing water and feed, the bacterial community structures seen 17 days after hatching were different. Moreover, although intestinal bacteria in the late stage of larvae were less diverse than those in the early stage of larvae, the specific bacteria were predominant, suggesting that the developed intestinal environment of the host puts selection pressure on the bacteria in the late stage. The specific bacteria in the late stage of larvae, which likely composed 'core microbiota', were also found on the egg surface. The present study highlights that proper management of the seed production process, including the preparation of rearing water, feeds, and fish eggs, is important for the aquaculture of healthy fish.

## Introduction

The intestinal microbiome of mammals has been well studied, and various findings have been obtained. In mammals, infancy and early childhood are the key periods for shaping the gut microbiome [1], which supports host functions such as indirect (immune system-mediated) and direct protection against pathogens [2] and nutrient absorption [3]. Hviid et al. [4] reported that the administration of antibiotics during this key period increases the incidence of other diseases, due to interference with the still unstable gut microbiome. Environmental factors play an important role in shaping the gut microbiome of mammals. The effects of oral acquisition are also important. For example, Ferretti et al. [5] highlighted that oral acquisition of bacteria from maternal skin, breast milk, feces, vagina, and oral cavity is crucial for the

aquaculture science of bluefin tuna and other cultured fish" from the Ministry of Education, Culture, Sports, Science and Technology of Japan (https://www.jsps.go.jp/english/index.html); JSPS KAKENHI Grant Number 25450286 (https://www.jsps.go.jp/english/index.html). The funders had no role in study design, data collection and analysis, decision to publish, or preparation of the manuscript.

**Competing interests:** The authors have declared that no competing interests exist.

development of infant microbiome. Furthermore, it has been shown that the gut microbiome of Japanese individuals exhibits a higher abundance of unique bacteria (*Bifidobacterium*) than that of individuals in other nations, indicating the influence of unique Japanese traditional foods [6].

The intestinal microbiome of fish has also been well studied. For example, Sugita et al. [7] using a direct counting method showed that the intestinal bacteria in eight marine fish species comprised approximately $10^9$–$10^{10}$ cells per g of intestinal contents. Such abundance of bacteria likely benefit the host fish as they do for mammals, improving digestion [8], production of vitamins [9], and exhibiting antibacterial activity against fish pathogens [10]. Environmental factors, among which food resources and habitat are important, shape the intestinal microbiome of fish [11–13]. Uchii et al. [11] reported that different feeding habitats among the same fish species led to the formation of different intestinal microbiota. The number of studies on fish intestinal bacteria has been increasing; however, their succession process during fish development from the egg is still unknown. This is due to the complex bacterial community in fish intestines and the surrounding environments such as water and feeds, which makes it difficult to trace the intestinal bacteria.

Pacific bluefin tuna *Thunnus orientalis* is one of the most popular and important fishery species, not only in Japan but also worldwide. Increasing global demand of bluefin tuna is threatening its natural population towards extinction. Therefore, various committees such as the Western and Central Pacific Fisheries Commission (WCPFC) have been discussing conservation and management of tuna resources every year. The Pacific bluefin tuna was reclassified from Vulnerable to Near Threatened in 2021 according to the International Union for Conservation of Nature (IUCN), but it is still under critical conditions. To remedy this alarming situation, Kindai University conducted full cycle aquaculture of bluefin tuna in 2002 [14]. Full-cycle aquaculture is an important technology that does not rely on natural resources and enables a stable supply of bluefin tuna, without threatening the natural population. However, the full cycle aquaculture of bluefin tuna also has various problems, such as disease and feeding regime during larval production [14,15].

The intestinal microbiota is important for the health and development of bluefin tuna, however, studies on this topic are limited [16,17]. Minich et al. [17] investigated the microbial diversity associated with mucosal membranes, including the gut farmed southern bluefin tuna (*Thunnus maccoyii*) and showed that the microbiota structure was affected by pontoon location and treatment with an antihelminthic drug. Gatesoupe et al. [16] investigated the change in intestinal microbiota in Atlantic bluefin tuna *Thunnus thynnus* larvae and showed that the microbiota varied greatly among individuals. Pathogenic bacteria such as *Photobacterium* and *Vibrio* species have been shown to cause disease in tuna [18–21]. Given that blood flukes often cause serious problems in bluefin tuna aquaculture [22] and also the 'pathobiome' concept [23], it is important to understand how intestinal bacteria directly and indirectly affect the health of the host.

In this study, we investigated the succession process of intestinal bacteria during the seed production in full cycle aquaculture of bluefin tuna. We hypothesized that bacteria derived from rearing water and feeds would colonize the intestinal tract of bluefin tuna, thereby shaping the gut bacterial community. In this study, we used healthy bluefin tuna larvae produced commercially at Susami Hatchery, Aquaculture Technology and Production Center, Kindai University.

## Materials and methods

### Sample collection

All experimental procedure was approved by the Institutional Animal Care and Use Committee and was conducted in accordance with the Kindai University Animal Experimentation

Regulations (2021-A-00118). This study was performed in accordance with ARRIVE guidelines. Informed consent was not required for this study.

Samples were collected three times: once in 2012 (expressed as Exp12) and twice in 2013 (expressed as Exp13r1 and Exp13r2). We used a full-cycle aquaculture of Pacific bluefin tuna *Thunnus orientalis*, rearing water, and feeds. The larval fish density was 150,000 fish/30 t tank in Exp12, and 250,000 and 500,000 fishes/50 t tank for Exp13r1 and Exp13r2, respectively. The periods were set from the egg to offshore for the analysis, and the feeding schedule was performed according to Sawada et al. [14] (S1 Fig and S1 Table).

The rearing water and feed were sampled immediately after feeding to the fish, and fish samples were taken immediately before feeding on the next day to investigate the relationship between intestinal and environmental bacteria. The fish samples were collected with a beaker or fry net, depending on the swimming capability. For the feed samples, rotifers (*Brachionus plicatilis*), *Artemia* (*Artemia salina*) nauplii, and feeder larvae (striped beakfish) were placed in a beaker, followed by collection onto a 60-μm-opening nylon membrane, and the commercial pellets were placed in a plastic bag. The rearing water pre-filtered, to remove feeds using a 60-μm-opening nylon membrane, was collected into a 50 mL centrifuge tube. All samples were immediately stored at −60°C until further analysis. The beaker, net, and nylon membranes were washed with hypochlorite.

For the egg and fish samples, we first performed a washing step to remove the bacteria from the samples. Water around the egg (71–120 eggs) and larval (16–44 fish; total length (TL) 2.0–4.6 mm) samples on the nylon membrane were removed with a 47 mm diameter GF/C filter (1822–047, Whatman, Kent, UK; pre-combusted at 450°C for 2 h) and washed with 5 mL of 0.22-μm-filtered seawater. This washing step was repeated seven times to prevent bacterial contamination from the rearing water. The whole fish body was subjected to DNA extraction because it was difficult to remove only the intestine. For experimental convenience, the bacteria of eggs and small fish were treated as intestinal bacteria. As for the larger larvae and fry (5–44 fish, TL 7.2–65.6 mm), the water around each individual fish was removed in the same manner as described above and washed with 1 mL of 0.22-μm-filtered seawater. This washing step was repeated seven times. The intestines of larvae and fry were removed with a sterilized razor blade and placed into a ZircoPrep Mini tube (FGM50M, NIPPON Genetics Co. Ltd., Tokyo, Japan) containing 400 μL of TE buffer (10 mM Tris-HCl, 1 mM EDTA, pH 8.0).

For the feed samples, rotifers, *Artemia*, and the feeder larvae of striped beakfish were washed in the same manner as the egg and fish samples. The weights of rotifer, *Artemia*, and feeder larvae were 33.2–53.5, 40.5–101.2, and 35.4–82.7 mg, respectively. The commercial pellets were weighed to approximately 50 mg. Each sample was placed in a ZircoPrep Mini tube containing 400 μL of TE buffer. For the water sample, 10 mL of water was filtered through a 0.2-μm-pore size polycarbonate membrane filter (16020002, Advantec Toyo Co. Ltd., Tokyo, Japan) under vacuum < 0.02 MPa. A 0.45-μm-pore size nitrocellulose membrane filter (HAWP02500, Millipore, MA, USA) was used as a base filter to prevent the contamination of filtered seawater. The polycarbonate filter was placed in a ZircoPrep Mini tube containing 400 μL of TE buffer. Filtered seawater was also subjected to filtration in the same manner to check for bacterial contamination.

## DNA extraction

The ZircoPrep Mini tubes containing the samples were vortexed for 15 min using a Vortex-Genie 2 (G-560, Scientific Industries, Inc., NY, USA). Lysozyme (final concentration 15 mg/mL) was added to the tubes and incubated for 60 min at 37°C with gentle mixing. After incubation, proteinase K (2 mg/mL) and SDS (1.8%) were added and incubated for 60 min at 55°C

with gentle mixing. After incubation, equal amounts of phenol were added to the supernatant and centrifuged at $20,630 \times g$ for 5 min at 20˚C. This phenol step was repeated three times. An equal amount of phenol/chloroform supernatant was added and centrifuged at $20,630 \times g$ for 5 min at 20˚C. Ethanol precipitation was subsequently performed. The DNA was dissolved in 30 μL of TE buffer, and the concentration was measured at 260 nm using a NanoDrop spectrophotometer (ND-1000, Thermo Fisher Scientific Inc., MA, USA).

## Automated ribosomal intergenic spacer analysis (ARISA)

PCR amplification was performed in a 20 μL reaction mixture using TaKaRa Ex Taq polymerase (Takara Bio Inc., Shiga, Japan) in a C1000 Thermal Cycler (Bio-Rad, Richmond, CA, USA). The mixture contained $1 \times$ PCR buffer, 2 mM $MgCl_2$, 0.2 mM of each dNTP, primers, 0.8 mg/mL bovine serum albumin, and 0.025 units TaKaRa Ex Taq polymerase. The primers used in this study were 16S-1392F (5□-G[C/T]A CAC ACC GCC CA-3′) and 23S-125R (5□-GGG TT[C/G/T] CCC CAT TC[A/G] G-3′) labeled with 6-FAM at the 5′ end [24]. The following PCR cycling conditions were used: initial denaturation step at 95˚C for 3 min, 30 cycles of 30 s at 95˚C, 30 s at 56˚C, and 45 s at 72˚C, followed by a final extension step at 72˚C for 7 min. PCR products were analyzed using 2.0% agarose gel electrophoresis.

PCR products were purified using a PCR Clean-Up Mini Kit (Favorgen, Ring-Tung, Taiwan), and the concentration was measured using a NanoDrop spectrophotometer. Purified products were diluted to 25 ng/μL with TE buffer, and 100 ng of a standardized amount were loaded into the fragment analysis. Products were then run for 3 h on an ABI 3130*xl* Genetic Analyzer (Applied Biosystems, Waltham, MA, USA) with a GeneScan 1200 LIZ dye Size Standard (Applied Biosystems, Waltham, MA, USA). The ARISA peak patterns were analyzed using PeakStudio version 2.2 [25]. According to Chow et al. [26], ARISA peaks were manually binned with maximum bin sizes of 1, 2, 3, and 5 bp for 390–450, 450–650, 650–900, and 900–1200 bp, respectively. Each ARISA peak was regarded as an operational taxonomic unit (OTU), and the relative abundance (peak height divided by the cumulative height of all peaks in the sample) was subjected to community analysis.

## Community analysis

All analyses were performed using R software [27]. Diversity indices (OTU number, logarithm of inverse Simpson and Shannon-Weiner indices) were calculated, using the 'Vegan' package [28]. Scatter plots of OTU abundance numbers and diversity indices during experiments were generated and the Pearson's coefficients were calculated using the 'ggpubr' package [29]. Bray-Curtis dissimilarities were calculated, and the distance matrix was analyzed using the between-group average linkage method for clustering and the 'Vegan' package. We performed a similarity profile (SIMPROF) test (9999 permutations with a $p < 0.0001$) to determine significant clusters among samples, using the 'clustsig' package [30]. Also, a principal coordinates analysis (PCoA) and an Adonis test (9999 permutations) were used to show OTU variations among samples, using 'ape' [31] and 'Vegan' [28] packages. Venn analysis was performed between the early and late stages of larvae as the result of cluster analysis, using the 'VennDiagram' package [32], to reveal characteristic OTUs. The derived OTUs and abundance were visualized by a heatmap using the 'tidyr' package [33].

## Results

### Bacterial diversity

During the experiments, bacterial OTU numbers of rearing water, feeds, and intestinal samples ranged from 7 to 81 OTUs, 2 to 80 OTUs, and 1 to 49 OTUs, respectively (Tables 1–3);

**Table 1. Diversity indices of bacterial communities in Exp12.**

| | Rearing water | | | | | | | | | | | | Feed | | | | | | | | | | | Intestine | | | | | | | | | | | |
|---|---|---|---|---|---|---|---|---|---|---|---|---|---|---|---|---|---|---|---|---|---|---|---|---|---|---|---|---|---|---|---|---|---|---|---|---|
| | W01 | W02 | W05 | W08 | W11 | W13 | W16 | W19 | W22 | W25 | W28 | W31 | R02 | R05 | R08 | R11 | A13 | AL16 | L19 | P19 | L22 | P25 | P28 | G-1 | G01 | G03 | G06 | G09 | G12 | G14 | G17 | G20 | G26 | G29 | G31 |
| OTU number | 11 | 42 | 36 | 42 | 28 | 40 | 41 | 40 | 36 | 32 | 19 | 39 | 57 | 80 | 33 | 17 | 18 | 29 | 36 | 38 | 45 | 34 | 31 | 10 | 8 | 34 | 10 | 25 | 23 | 15 | 9 | 7 | 6 | 6 | 6 |
| $H'$ | 3.02 | 4.10 | 4.48 | 4.11 | 3.97 | 4.36 | 4.29 | 4.24 | 3.94 | 3.91 | 3.24 | 3.06 | 5.05 | 5.24 | 4.04 | 1.94 | 3.30 | 3.21 | 4.28 | 3.41 | 4.38 | 3.49 | 2.97 | 3.07 | 2.29 | 3.16 | 3.28 | 4.26 | 3.62 | 3.52 | 2.98 | 2.57 | 2.50 | 2.20 | 2.41 |
| $\log(1/D)$ | 2.73 | 3.43 | 4.08 | 3.34 | 3.46 | 3.78 | 3.75 | 3.43 | 2.98 | 2.84 | 2.50 | 1.89 | 4.47 | 4.42 | 3.44 | 1.23 | 2.55 | 2.47 | 3.74 | 2.39 | 3.81 | 2.47 | 1.76 | 2.85 | 1.92 | 1.74 | 3.25 | 3.98 | 3.00 | 3.16 | 2.82 | 2.42 | 2.42 | 1.99 | 2.30 |

W, water; R, rotifer; A, *Artemia*; L, feeder larvae; P, commercial pellet; G, intestine. The number after letters indicates the day after hatching of bluefin tuna. Day 0 is the day of hatching and Day −1 represents the unhatched egg.

**Table 2. Diversity indices of bacterial communities in Exp13r1.**

| | Rearing water | | | | | | | | | Feed | | | | | | | Intestine | | | | | | | |
|---|---|---|---|---|---|---|---|---|---|---|---|---|---|---|---|---|---|---|---|---|---|---|---|---|
| | W-1 | W01 | W03 | W07 | W11 | W13 | W15 | W19 | W25 | R03 | A07 | A11 | L14 | AL15 | L16 | L19 | G01 | G03 | G04 | G08 | G12 | G16 | G25 | G35 |
| OTU number | 61 | 7 | 29 | 27 | 60 | 18 | 22 | 27 | 14 | 5 | 42 | 21 | 63 | 64 | 25 | 7 | 49 | 45 | 17 | 19 | 1 | 1 | 6 | 9 |
| $H'$ | 5.17 | 2.57 | 3.70 | 3.40 | 4.17 | 3.31 | 3.34 | 3.67 | 2.93 | 1.46 | 3.62 | 3.45 | 4.32 | 5.04 | 3.64 | 1.32 | 3.54 | 4.68 | 2.85 | 2.71 | - | - | 2.21 | 2.69 |
| $\log(1/D)$ | 4.41 | 2.40 | 2.99 | 2.48 | 3.25 | 2.98 | 2.76 | 3.16 | 2.36 | 0.95 | 2.48 | 2.85 | 3.35 | 4.50 | 2.83 | 0.78 | 2.58 | 4.04 | 2.28 | 2.13 | - | - | 2.03 | 2.41 |

W, water; R, rotifer; A, *Artemia*; L, feeder larvae; P, commercial pellet; G, intestine. The number after letters indicates the day after hatching of bluefin tuna. Day 0 is the day of hatching and Day -1 represents the unhatched egg.

however, no PCR products were obtained from some intestinal samples (S1 Table). The number of OTUs in the rearing water sample tended to be smaller before the beginning of feeding, and those of feed samples varied, even though the feed type was the same. The OTU numbers of intestinal samples decreased with the progression of larval growth (Pearson's correlation coefficient $r = -0.498$, $n = 27$, $p < 0.01$), unlike those of seawater ($r = 0.394$, $n = 31$, $p < 0.05$) (S2 Fig). Simpson and Shannon-Wiener indices of rearing water sample ranged from 1.42 to 4.47 and 1.88 to 5.17, respectively, and those of feed samples ranged from 0.27 to 4.50 and 0.46 to 5.24, respectively. In intestinal samples, Simpson and Shannon-Wiener indices ranged from 0.08 to 4.04 and 0.23 to 4.68, respectively. No correlation of the indices with the progression of larval growth was observed for all samples of rearing water, feed, or intestine.

## Bacterial community structure

ARISA peak patterns were separated into two clusters, communities of rearing water and feed, regardless of the sampling year (Figs 1A and S3A). The early and late stages of intestinal communities appeared to resemble communities of rearing water and feed, respectively. Examining the communities of intestinal samples in detail, 17 days after hatching (DAH) formed the same cluster ($p < 0.0001$ SIMPROF and Adonis tests, Cluster-L, Figs 1B and S3B). Venn analysis between Cluster-L and the other clusters (Cluster-E) from the cluster analysis revealed characteristic OTUs: 130 in Cluster-E, 4 in Cluster-L, and 13 in both clusters (Fig 2 and S2 Table in S1 File). Although most of the OTUs were observed in the early stage of larvae (Cluster-E), the OTUs of intestinal samples in the late stage (Cluster-L) consisted of a small number of OTUs (Cluster-L ∩ Cluster-E). In response to this result, the occurrence pattern and abundance variation of characteristic OTUs in both clusters (Cluster-L ∩ Cluster-E) and in only Cluster-L (Cluster-L \ Cluster-E) were visualized by heatmap analysis (Fig 3). Regardless of the year of the experiments, six OTUs (OTU507.9, OTU553.3, OTU591.1, OTU653.1, OTU789.5, and OTU814.4) were particularly predominant and frequently observed in intestinal samples. In some cases (e.g., 12_G26 in Fig 3), intestinal samples contained only these six OTUs. Among them, OTU507.9, OTU591.1, OTU653.1, and OTU789.5 were detected throughout the experiment and became dominant after 17 DAH. Notably, these OTUs were also detected in the egg samples (12_G-1 and 13r2_G-1 in Fig 3). OTU553.3 and OTU814.4 were less dominant in the early stage, but more frequently observed after 17 DAH. Most of the other OTUs observed in the rearing water and feed samples were not predominant in the intestinal samples. The OTU421.9 was observed frequently and predominantly in the samples of rearing water and feeds and the intestinal samples before 17 DAH. However, it was seldom observed in the intestinal samples after 17 DAH. Four OTUs (OTU420.5, OTU555.0, OTU622.9, and OTU669.4) in Cluster-L (Cluster-L \ Cluster-E) were detected but were not

**Table 3. Diversity indices of bacterial communities in Exp13r2.**

| | Rearing water | | | | | | | | | | Feed | | | | | | | | Intestine | | | | | | |
|---|---|---|---|---|---|---|---|---|---|---|---|---|---|---|---|---|---|---|---|---|---|---|---|---|---|
| | W-1 | W01 | W02 | W05 | W08 | W12 | W16 | W20 | W24 | W29 | R01 | R02 | R05 | R08 | R12 | L13 | L16 | L20 | G-1 | G00 | G03 | G06 | G21 | G25 | G29 |
| OTU number | 7 | 9 | 9 | 15 | 10 | 44 | 56 | 58 | 58 | 81 | 10 | 12 | 50 | 28 | 15 | 2 | 20 | 18 | 33 | 13 | 4 | 2 | 3 | 7 | 8 |
| H' | 1.88 | 2.07 | 2.65 | 2.36 | 2.28 | 3.94 | 4.18 | 3.91 | 4.18 | 5.17 | 2.27 | 2.75 | 3.94 | 2.81 | 3.56 | 0.46 | 3.39 | 3.62 | 4.39 | 2.72 | 0.23 | 0.98 | 1.58 | 2.41 | 2.57 |
| log(1/D) | 1.42 | 1.57 | 2.41 | 1.47 | 1.63 | 3.12 | 3.32 | 3.10 | 3.40 | 4.47 | 1.74 | 2.33 | 3.16 | 2.08 | 3.30 | 0.27 | 2.95 | 3.29 | 3.78 | 2.11 | 0.08 | 0.96 | 1.57 | 2.10 | 2.34 |

W, water; R, rotifer; A, *Artemia*; L, feeder larvae; P, commercial pellet; G, intestine. The number after letters indicates the day after hatching of bluefin tuna. Day 0 is the day of hatching and Day -1 represents the unhatched egg.

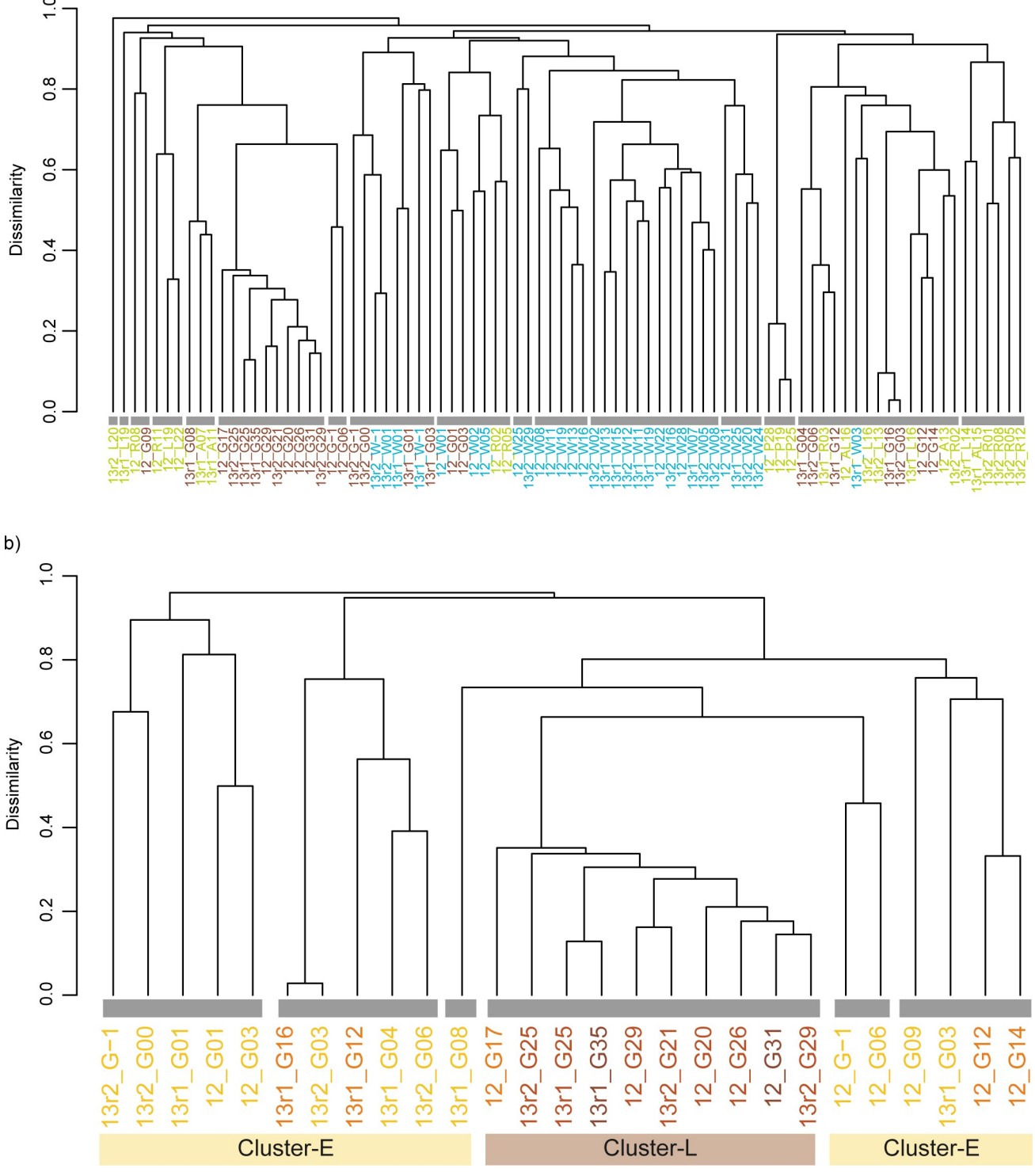

**Fig 1. Cluster analysis of bacterial community structures.** (a) Cluster constructed using all samples, rearing water (blue letters), feed (green letters) and intestine (brown letters). (b) Cluster constructed using only intestinal samples. Sample names are represented by experimental year, sample type, and DAH of bluefin tuna; 0 DAH is the day of hatching and Day -1 represents the unhatched egg. For example, 12_G17 represents intestinal samples at 17 DAH in 2012. W, water; G, intestine; R, rotifer, A, Artemia; L, feeder larvae (striped beakfish); P, commercial pellet.

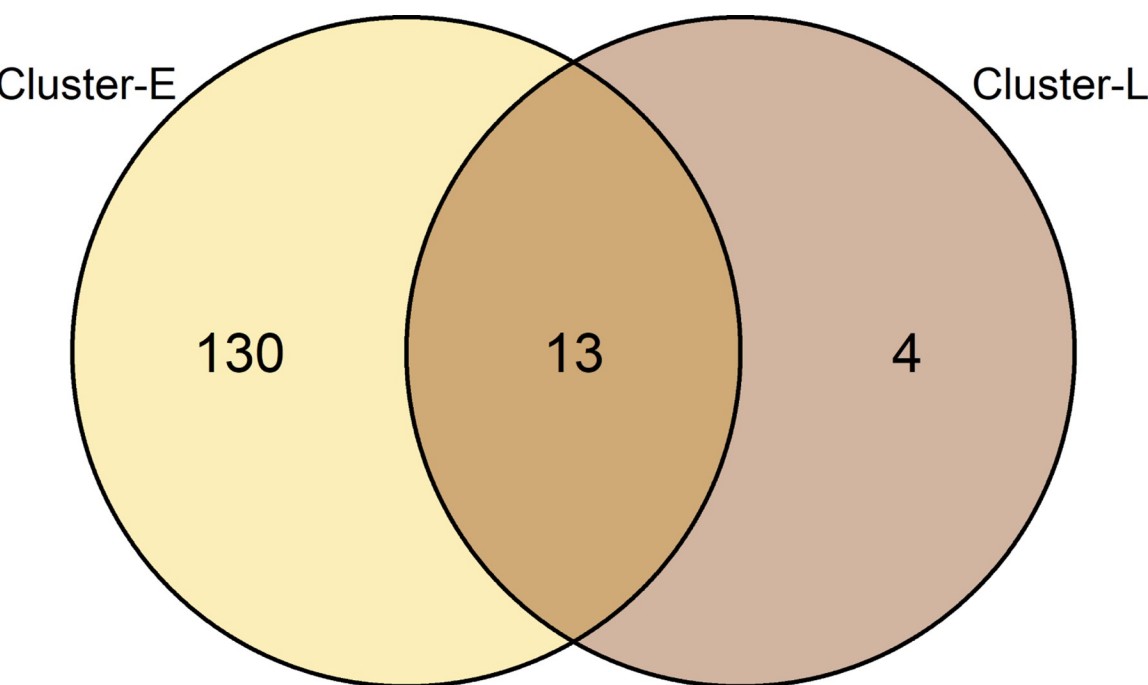

**Fig 2. Venn diagram of OTUs in Cluster-E and Cluster-L from cluster analysis.** The name of clusters corresponds to descriptions in Figs 1B and S3B.

predominant throughout the experiment in the intestinal samples as well as in the rearing water and feed samples. The distribution of the other OTUs (Cluster-E \ Cluster-L) is shown in S4 Fig.

## Discussion

In the present study, we demonstrated the succession process of intestinal bacteria in bluefin tuna, in full-cycle aquaculture for the first time. For the analysis, we used a PCR-based finger-printing analysis, ARISA, which seemed to detect particularly the dominant bacteria in a sample, with some bias [34]. Kashinskaya et al. [35] used several molecular methods to compare differences in the intestinal bacterial communities of Prussian carp and reported that the PCR-based cloning method successfully appeared to detect dominant bacteria in the samples. It is also reported that there is ecological coherence of bacterial diversity pattern between ARISA and deep sequencing techniques [36–38]. Therefore, we intended to see a similar trend using the PCR-based method, ARISA, in this study. The dominant bacteria may have a great impact on host development, as can be seen from the probiotic strategy. The simplification in the present study would be key to revealing the succession process of intestinal microbiota. In addition, bacterial contamination from the body surface might be observed in samples of the smaller larvae due to the handling of the whole fish body. Yoshimizu et al. [39] detected bacterial counts in the range of $10^2$–$10^3$ colony forming units/g of body weight from the body surface of the sac fry. However, the contamination risk could be significantly low in this study as our washing steps were notably more frequent (at least $10^7$ times higher) than those of Yoshimizu et al. [39]. Our experiments yield three major findings: 1) the intestinal bacteria in the early stage of larvae were affected by rearing water and feed, 2) the specific intestinal bacteria in the late stage of larvae became dominant regardless of the diversity of rearing water and

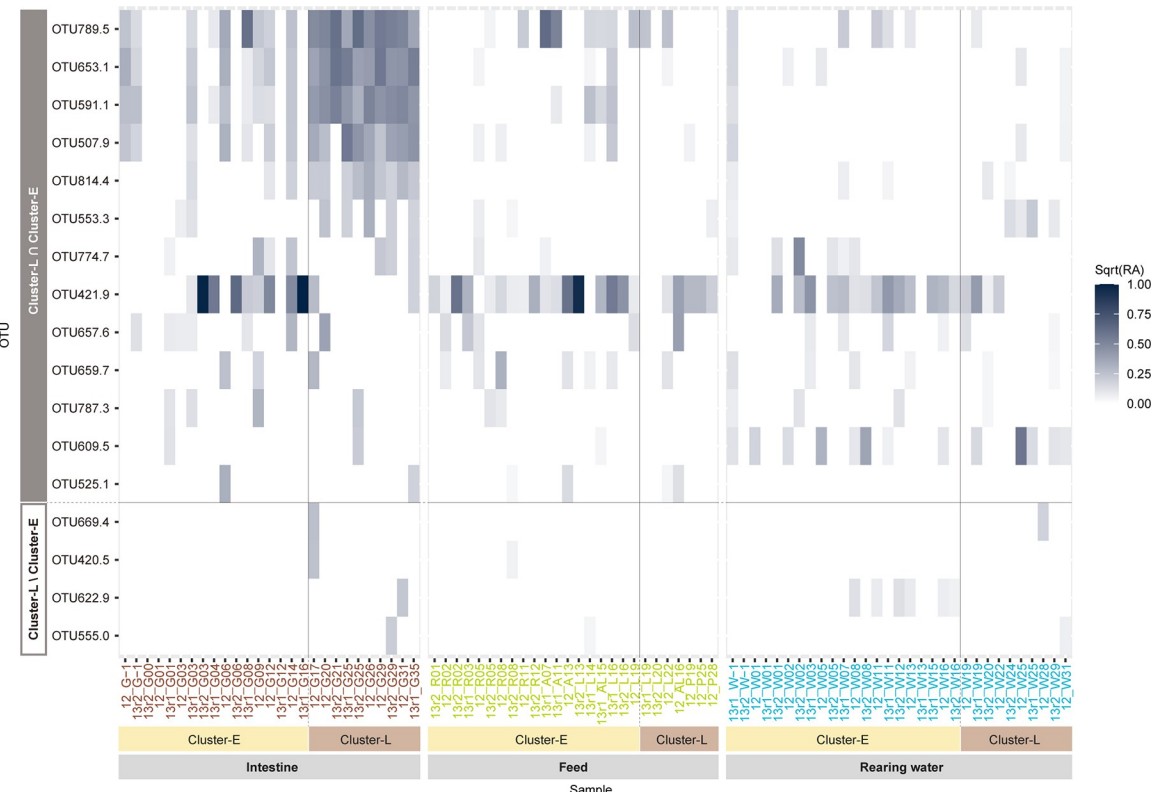

**Fig 3. Heatmap analysis of characteristic OTUs derived from Venn analysis.** The heatmap color corresponds to the square root of transformed relative abundance of each OTU in each sample. Sample names are represented by experimental year, sample type (rearing water, blue letters; feeds, green letters; intestine, brown letters), and DAH of bluefin tuna; 0 DAH is the day of hatching and Day -1 represents the unhatched egg. For example, 12_G17 represents intestinal samples of 17 DAH in 2012. The order of samples is based on the hatching day of larvae. W, water; G, intestine; R, rotifer; A, Artemia; L, feeder larvae (striped beakfish); P, commercial pellet.

feed, and 3) the specific intestinal bacteria in the late stage of larvae, especially after 17 DAH, originated from the egg surface.

This study suggests that the management of rearing water and live feeds, such as rotifers, is important for controlling the bacterial communities in seed production because they affect the intestinal bacteria in the early stage of larvae (Figs 1A and S3A). Other studies have also shown that fish intestinal bacteria are mainly affected by feed [12,40–43], water [44,45], and both [39,46–48]. Various types of water used for seed production have different bacterial communities [49]. They mentioned that the management of various waters in seed production was a key point because pathogenic microbes would increase during larval development even though the initial rearing water was free of pathogens. Effective management of rearing water has been shown to be 'green-water', a type of rearing water containing a large amount of microalga [50–53]. Studies have shown that live microalgae such as *Nannochloropsis oculata* and *Chlorella vulgaris* could eradicate the pathogenic bacterium *Vibrio anguillarum* by collaborating with the indigenous probiotic bacterium *Sulfitobacter* in rearing water. This 'green-water' technique is necessary to maintain the bacterial communities in desirable condition in the early stage of larvae.

Specific bacteria in the late stage of larvae in this study would colonize the favorable intestinal environment of bluefin tuna (Figs 1B and S3B) by the host and/or interactions of host-microbe and microbe-microbe [39,46–48,54–56]. A pioneering study of the succession process

of intestinal bacteria by Yoshimizu et al. [39] indicated the possibility of regulation by the host. They reported that intestinal bacteria in the early stage of salmonid larvae were affected by the rearing water and diet, and then the developed intestinal environment by the activation of digestive tract would establish the characteristic intestinal microbiota. Ontogenetic development of the digestive tract and enzyme activity in bluefin tuna has been reported to develop relatively fast [57,58]. In bluefin tuna farmed at Kindai University, Miyashita et al. [59] showed that activities of pepsin-like and trypsin-like enzymes increased simultaneously with the development of stomach and pyloric caecum functions during the transitional period of juvenile tuna (17–25 DAH) as the rate of percentage of preanal length to standard length increased. The selection pressure would be due to host regulation as well as host-microbe and microbe-microbe interactions [55]. Changes in microbiota contribute to larval health and growth by improving host abilities.

The possible source of characteristic intestinal bacteria in the late stage of larvae would be the egg surface where these characteristic bacteria were also found (Fig 3), and the bacteria should compose of 'core microbiota' [60,61] in bluefin tuna. Fish eggs contain many bacteria on their surfaces [39,41,62], although sometimes the associated bacteria can cause mortality in marine fish [63]. Lauzon et al. [64] demonstrated that probiotic bathing treatment of eggs resulted in the establishment of added bacteria in the larval intestine, leading to increased survival, stress tolerance, and growth of host larvae. Notably, they highlighted that bacterial control was mainly evidenced prior to larval feeding, suggesting the importance of bacterial establishment on the egg surface. The mechanism of transfer from egg surface to larva is still unknown in bluefin tuna, but one possibility is ingestion of egg surface bacteria via grazing on egg debris and/or drinking seawater [62,65]. In the present study, we demonstrated that bacteria on the egg surface would be an important key species for determining 'core microbiota' of intestine in bluefin tuna. As bacteria associated with fish egg surface are likely to be opportunistic species [63], the management of eggs shortly after release, including disinfection [15] and preparation of rearing water, would be an important process. Further investigation is needed to determine whether the key species are still colonized in the adult bluefin tuna.

In conclusion, a clear succession process of intestinal bacteria in full-cycle aquaculture of Pacific bluefin tuna was shown for a total of three times of seed production. While bacteria in the early stage of larvae were affected by bacteria of rearing water and feeds, in the late stage (especially, after 17 DAH when the intestinal environment was developed), they changed to specific bacteria originating from the egg surface, which are likely to be composed of the 'core microbiota'. Thus, we highlighted the importance of proper management in the seed production process, including egg management for the aquaculture of healthy fish. Further research is needed to determine the bacterial function and whether the bacteria are well colonized in adult fish.

## Supporting information

**S1 Fig. Growth curve of Pacific bluefin tuna and feeding schedules.** The growth of bluefin tuna is indicated as a plot graph, and the feeding schedules are shown as a bar graph at the top of the plot graph. Exp12, Exp13r1, and Exp13r2 indicate the year (2012 or 2013) and round (r1 or r2) in which the experiment was performed.
(TIF)

**S2 Fig. Variation of OTU abundance and diversity indices during experiments.** a)–c) Show the number of OTUs in intestine, feed, and water samples, respectively. d)–f) Show indices of the Shannon-Wiener (*H'*) (open circle) and Simpson (log(1/*D*)) (filled orange circle) in

intestine, feed, and water samples, respectively. *r* and p values describe Pearson's correlation.
(TIF)

**S3 Fig. Principal coordinate analysis (PCoA) of bacterial community structures.** (a) PCoA constructed using all samples, rearing water (blue letters), feed (green letters) and intestine (brown letters). (b) PCoA constructed using only intestinal samples, the early (yellow group) and late (brown group) stages based on Adonis test ($p = 0.0001$). Sample names are represented by experimental year, sample type, and DAH of bluefin tuna; 0 DAH is the day of hatching and Day -1 represents the unhatched egg. For example, 12_G17 represents intestinal samples of 17 DAH in 2012. W, water; G, intestine; R, rotifer, A, Artemia; L, feeder larvae (striped beakfish); P, commercial pellet.
(TIF)

**S4 Fig. Heatmap analysis of all intestinal OTUs.** The heatmap color corresponds to the square root of transformed relative abundance of each OTU in each sample. Sample names are represented by experimental year, sample type (rearing water, blue letters; feeds, green letters; intestine, brown letters), and DAH of bluefin tuna; 0 DAH is the day of hatching and Day -1 represents the unhatched egg. For example, 12_G17 represents intestinal samples of 17 DAH in 2012. The order of sample is based on the hatching day of larvae. W, water; G, intestine; R, rotifer, A, Artemia; L, feeder larvae (striped beakfish); P, commercial pellet.
(TIF)

**S1 Table. Sample list in this study.**
(XLSX)

**S1 File. Relative abundance of all OTUs in this study.**
(XLSX)

## Acknowledgments

We would like to thank Amal Biswas and Yasuo Agawa for their valuable suggestions. We would also like to thank Editage (www.editage.com) for English language editing. We sincerely appreciate two anonymous reviewers for their helpful suggestions.

## Author Contributions

**Conceptualization:** Akito Taniguchi, Ryuichiro Aoki, Mitsuru Eguchi.

**Data curation:** Akito Taniguchi, Ryuichiro Aoki.

**Formal analysis:** Akito Taniguchi, Ryuichiro Aoki.

**Funding acquisition:** Mitsuru Eguchi.

**Investigation:** Akito Taniguchi, Ryuichiro Aoki.

**Methodology:** Akito Taniguchi, Ryuichiro Aoki.

**Project administration:** Akito Taniguchi, Mitsuru Eguchi.

**Resources:** Isamu Inoue.

**Supervision:** Mitsuru Eguchi.

**Visualization:** Akito Taniguchi, Ryuichiro Aoki.

**Writing – original draft:** Akito Taniguchi, Ryuichiro Aoki.

**Writing – review & editing:** Akito Taniguchi, Ryuichiro Aoki, Isamu Inoue, Mitsuru Eguchi.

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
