## [Decision Letter · Decision Letter 0]

13 May 2022

PONE-D-21-35860Succession of the intestinal bacterial community in Pacific bluefin tuna (Thunnus orientalis) larvaePLOS ONE

Dear Dr. Eguchi,

Thank you for submitting your manuscript to PLOS ONE. After careful consideration, we feel that it has merit but does not fully meet PLOS ONE’s publication criteria as it currently stands. Therefore, we invite you to submit a revised version of the manuscript that addresses the points raised during the review process.

1. The data cannot support the conclusions. PLOS ONE is designed to communicate primary scientific research, and welcome submissions in any applied discipline that will contribute to the base of scientific knowledge. But the data of this manuscript cannot support the conclusions.

2. This manuscript has the statistical analysis problem.

3. The revised manuscript needs to address each of the comments of the reviewers.

We look forward to receiving your revised manuscript.

Kind regards,

Tzong-Yueh Chen, Ph.D.

Academic Editor

PLOS ONE

**Journal requirements:**

2. PLOS requires an ORCID iD for the corresponding author in Editorial Manager on papers submitted after December 6th, 2016. Please ensure that you have an ORCID iD and that it is validated in Editorial Manager. To do this, go to ‘Update my Information’ (in the upper left-hand corner of the main menu), and click on the Fetch/Validate link next to the ORCID field. This will take you to the ORCID site and allow you to create a new iD or authenticate a pre-existing iD in Editorial Manager. Please see the following video for instructions on linking an ORCID iD to your Editorial Manager account: https://www.youtube.com/watch?v=_xcclfuvtxQ.

**Reviewers' comments:**

Reviewer's Responses to Questions

**Comments to the Author**

1. Is the manuscript technically sound, and do the data support the conclusions?

Reviewer #1: Partly

Reviewer #2: Partly

2. Has the statistical analysis been performed appropriately and rigorously? 

Reviewer #1: N/A

Reviewer #2: I Don't Know

3. Have the authors made all data underlying the findings in their manuscript fully available?

Reviewer #1: Yes

Reviewer #2: Yes

4. Is the manuscript presented in an intelligible fashion and written in standard English?

Reviewer #1: Yes

Reviewer #2: Yes

5. Review Comments to the Author

Reviewer #1: The author examined the gut microbiota of blue fin tuna from the offshore cage rearing system by Automated approach for ribosomal intergenic spacer (ARISA) analysis. However, even with dissimilarity tree analysis, this method can give only preliminary dominant microbiota change but not enough to state the solid succession limited by the lack of bacteria annotation and the multi-analysis of the microbiota change of feeds and rearing water due to seemingly technical issue of DNA extraction and the uneven distribution of bacteria abundance.

Major issue:

1. The biggest issue with this work is still the accuracy of ARISA. Have author(s) done any single 16s for getting idea on whether How different OTU number or major OTU number difference? Even though the author(s) claimed ARISA can represent major dominant bacteria?

2. Figure 1. Can author(s) use PCoA to represent the beta-diversity of dominant bacteria from ARISA? PCoA can represent better than tree in terms of distinguishing the different microbiota group along with statistical analysis such as ANOSIM/ADONIS.

3. Venn diagram: how were the 117 and 4 distinguished bacteria distributed in the cluster E and L samples respectively?

4. It seems the type of feeding organisms changed during the seedling (Table. S1) and their microbiota could change (Fig. S2e). Can authors show how different the microbiota is between different feeding organisms? This will tell us how if the feeds can really impact the intestinal microbiota.

5. Figure 3. It seems a lot of OTUs (from 789.5- 774.7) were missing in the most of the rearing water and feed samples but presenting in either egg or larvae intestine. If so, then host factor can be more critical than feed/rearing water in microbiota succession?

6. Line179-180. however, no PCR products were obtained from some intestinal samples. Could it be the issue of bacteria DNA extraction? if so, then this will also affect the diversity and total abundance of the microbiota.

Minor issue:

1. Figure S2. – Y-axis. Number of OTU instead of Number if OTU. P value of R2?

Reviewer #2: The manuscript entitled “Succession of the intestinal bacterial community in Pacific bluefin tuna (Thunnus orientalis) larvae” is try to investigated the succession process of intestinal bacteria during larvae rearing and their relationship between rearing water and feed (rotifer, artemia, feeder larvae and commercial pellets. The bacterial community was detected by an “automated ribosomal intergenic spacer analysis (ARISA)” methods and the operational taxonomic unit (OTUs), logarithm of inverse Simpson and Shannon-Weiner indices were used as the diversity indices. I have to say that this experiment is difficult to perform due to the sample collection was difficult. This manuscript also may give us very different results. However, there are some issues still need to be clarified before this draft can be published.

1. Since the fish samples were collected individually, how to define the data collected from these fish is succession?

2. I’m curious what is the OTUs number variation between each pooled sample?

3. In the manuscript, the authors mentioned the bacterial community in fish larvae have great different in the late larvae stage (after 17 dph). Even the number of OTUs seems reduce after this stage. But how to define this observation.

4. As I understand, the next generation sequencing (NGS) methods had also can using in analyse bacterial community composition in gut or environment for some years. Not only give you the different OTUs reactions, but also can give you their composition. How you compare the ARISA and NGS methods using in bacterial community ananlysis in your research.

6. PLOS authors have the option to publish the peer review history of their article (what does this mean?). If published, this will include your full peer review and any attached files.

Reviewer #1: No

Reviewer #2: No

---

## [Author Response · Author response to Decision Letter 0]

25 Jun 2022

PONE-D-21-35860

Succession of the intestinal bacterial community in Pacific bluefin tuna (Thunnus orientalis) larvae

PLOS ONE

1. The data cannot support the conclusions. PLOS ONE is designed to communicate primary scientific research, and welcome submissions in any applied discipline that will contribute to the base of scientific knowledge. But the data of this manuscript cannot support the conclusions.

> This paper presents a molecular ecological approach to the succession pattern of intestinal bacteria. We did not perform species identification because the main purpose of the present manuscript is to understand how environmental factors such as feed and rearing water influence intestinal microbiota. We hope this meets the aim of your journal.

2. This manuscript has the statistical analysis problem.

> We have addressed this problem.

3. The revised manuscript needs to address each of the comments of the reviewers.

> We have responded to the reviewers’ comments individually.

Reviewers' comments:

Reviewer's Responses to Questions

Comments to the Author

1. Is the manuscript technically sound, and do the data support the conclusions?

Reviewer #1: Partly

Reviewer #2: Partly

2. Has the statistical analysis been performed appropriately and rigorously?

Reviewer #1: N/A

Reviewer #2: I Don't Know

3. Have the authors made all data underlying the findings in their manuscript fully available?

Reviewer #1: Yes

Reviewer #2: Yes

4. Is the manuscript presented in an intelligible fashion and written in standard English?

Reviewer #1: Yes

Reviewer #2: Yes

5. Review Comments to the Author

Reviewer #1: The author examined the gut microbiota of blue fin tuna from the offshore cage rearing system by Automated approach for ribosomal intergenic spacer (ARISA) analysis. However, even with dissimilarity tree analysis, this method can give only preliminary dominant microbiota change but not enough to state the solid succession limited by the lack of bacteria annotation and the multi-analysis of the microbiota change of feeds and rearing water due to seemingly technical issue of DNA extraction and the uneven distribution of bacteria abundance.

Major issue:

1. The biggest issue with this work is still the accuracy of ARISA. Have author(s) done any single 16s for getting idea on whether How different OTU number or major OTU number difference? Even though the author(s) claimed ARISA can represent major dominant bacteria?

> In this manuscript, we aimed to investigate how intestinal microbiota communities are formed with respect to microbes found in food and water. Therefore, we used ARISA, which is reported to be able to estimate microbial diversity and community composition in terms of OTUs (Fisher and Triplett, 1999). It is also a semi-quantitative technique (Brown et al, 2005) and has been utilized in many studies. In humans, it has been reported that the predominant bacteria have a significant impact on the host (Gerritsen et al, 2011), so understanding the microbial dynamics detected by ARISA will be very meaningful. However, as you pointed out, because species composition needs to be elucidated to utilize the data, a deep sequencing approach should be used in the future.

2. Figure 1. Can author(s) use PCoA to represent the beta-diversity of dominant bacteria from ARISA? PCoA can represent better than tree in terms of distinguishing the different microbiota group along with statistical analysis such as ANOSIM/ADONIS.

> Thank you for your suggestions. We performed PCoA and ADONIS (9999 permutations). These results are shown in supplementary Fig S3 because the eigenvalues of axes 1 & 2 were low, especially in PCoA of all samples (S3a Fig). The ADONIS results are also shown in Fig S3, which shows the adjusted R2 of 0.117 with a p value of 0.0001 among sample types, and an adjusted R2 of 0.213 with a p value of 0.0001 between Clusters E and L, indicating significant differences. Additionally, for the significance in the cluster analysis (Fig. 1), we performed SIMPROF analysis (9999 permutations with a p < 0.0001 according to ADONIS result). The results are shown in Fig. 1 as bars below each cluster. These results were similar to those yielded by PCoA and ADONIS.

3. Venn diagram: how were the 117 and 4 distinguished bacteria distributed in the cluster E and L samples respectively?

> Thank you for pointing this out. First, after rechecking all OTUs, we re-drew the Venn diagram (Fig. 2) because the OTU number detected from Cluster-E was wrong in the previous manuscript; the correct number was 130 OTUs, not 117 OTUs. The distribution of the 130 and 4 bacterial OTUs is shown in Fig S4.

4. It seems the type of feeding organisms changed during the seedling (Table. S1) and their microbiota could change (Fig. S2e). Can authors show how different the microbiota is between different feeding organisms? This will tell us how if the feeds can really impact the intestinal microbiota.

> Thank you for your comment. We showed the distribution in S4 Fig. Bacteria in feed sample are highly variable, but similar for the same feed items. Interestingly, especially during the feeding period of rotifers and Artemia, bacteria from them were detected as intestinal bacteria, but most of the bacteria were no longer detected after the period.

5. Figure 3. It seems a lot of OTUs (from 789.5- 774.7) were missing in the most of the rearing water and feed samples but presenting in either egg or larvae intestine. If so, then host factor can be more critical than feed/rearing water in microbiota succession?

> As you mentioned, we speculate that host factors, such as digestive enzymatic activity, are among the critical factors for determining microbiota composition. In this manuscript, we aimed to identify the general process of intestinal microbiota development, that is, to determine whether the microbiota come from water, feed items, or another source. 

6. Line179-180. however, no PCR products were obtained from some intestinal samples. Could it be the issue of bacteria DNA extraction? if so, then this will also affect the diversity and total abundance of the microbiota.

> Unfortunately, DNA extraction did not work for some samples. However, the method in this manuscript (physical disruption by bead beating; chemical lysis by lysozyme, proteinase K and SDS; and phenol/chloroform purification) has been used for DNA extraction in many previous papers. Every step commonly reported for effective DNA extraction from intestinal samples (e.g., Daly et al., 2012; Yuan et al., 2012; Eun et al., 2014; Dubin et al., 2016; Han et al., 2018). One of the reasons why no PCR products were obtained from some intestinal samples could be low abundance of intestinal microbes during development of host digestive tract. In any case, as you mentioned, care should be taken in interpreting the results because DNA extraction could introduce bias with regard to microbial diversity.

Minor issue:

1. Figure S2. – Y-axis. Number of OTU instead of Number if OTU. P value of R2?

> Thank you for pointing this out. We have corrected “if” to “of” and showed P- values in S2 Fig.

Reviewer #2: The manuscript entitled “Succession of the intestinal bacterial community in Pacific bluefin tuna (Thunnus orientalis) larvae” is try to investigated the succession process of intestinal bacteria during larvae rearing and their relationship between rearing water and feed (rotifer, artemia, feeder larvae and commercial pellets. The bacterial community was detected by an “automated ribosomal intergenic spacer analysis (ARISA)” methods and the operational taxonomic unit (OTUs), logarithm of inverse Simpson and Shannon-Weiner indices were used as the diversity indices. I have to say that this experiment is difficult to perform due to the sample collection was difficult. This manuscript also may give us very different results. However, there are some issues still need to be clarified before this draft can be published.

> Thank you for your comments.

1. Since the fish samples were collected individually, how to define the data collected from these fish is succession?

> The fish were reared under proper management, such as water quality, feed quality and quantity, and the number of reared fish from hatching to net-cage transfer. Fish were maintained under the same physiological conditions in each trial. In this manuscript, the formation of similar intestinal microbiota in the three rounds was considered evidence of this. We also collected multiple fish for each sample to decrease the effect of individual differences. Therefore, although fish samples were collected individually, we feel that our data are sufficient to indicate succession.

2. I’m curious what is the OTUs number variation between each pooled sample?

> “Bacterial abundance in rearing water, feed, and intestine samples ranged from 7–81, 2 –80, and 1– 49 OTUs, respectively (Tables 1–3)” lines 195–196 in “Revised Manuscript with Tracked Change”. This variation likely occurred because the predominant bacteria changed during the development of the host gastrointestinal tract. In this manuscript, we standardized the DNA amount loaded (100 ng) into the ARISA instrument to reduce the variation between samples (line 168). 

3. In the manuscript, the authors mentioned the bacterial community in fish larvae have great different in the late larvae stage (after 17 dph). Even the number of OTUs seems reduce after this stage. But how to define this observation.

> This probably occurs due to the development of the host gastrointestinal tract, particularly the increased activities of digestive enzymes. Only bacteria that can tolerate the intestinal environment can survive. 

4. As I understand, the next generation sequencing (NGS) methods had also can using in analyse bacterial community composition in gut or environment for some years. Not only give you the different OTUs reactions, but also can give you their composition. How you compare the ARISA and NGS methods using in bacterial community ananlysis in your research.

> As you mentioned, the NGS method is one of the strongest approaches for analysis of intestinal microbiota. However, in this manuscript, we focused on the relationship between the intestinal bacteria and the bacteria found in the feed and rearing water, which does not necessarily require identification of bacterial species composition. In fact, using ARISA, which is more economical than NGS, it is easy to compare multiple samples. We identified the importance of egg management for the formation of intestinal microbiota in the late stage. When compared to NGS data, the trend in major species may not change because ARISA detects predominant bacteria. However, because NGS can detect rare species, it may provide different findings in this respect.

6. PLOS authors have the option to publish the peer review history of their article (what does this mean?). If published, this will include your full peer review and any attached files.

Do you want your identity to be public for this peer review? For information about this choice, including consent withdrawal, please see our Privacy Policy.

Reviewer #1: No

Reviewer #2: No

---

## [Decision Letter · Decision Letter 1]

9 Aug 2022

PONE-D-21-35860R1Succession of the intestinal bacterial community in Pacific bluefin tuna (Thunnus orientalis) larvaePLOS ONE

Dear Dr. Eguchi,

Thank you for submitting your manuscript to PLOS ONE. After careful consideration, we feel that it has merit but does not fully meet PLOS ONE’s publication criteria as it currently stands. Therefore, we invite you to submit a revised version of the manuscript that addresses the points raised during the review process.

1. The data cannot support the conclusions. PLOS ONE is designed to communicate primary scientific research, and welcome submissions in any applied discipline that will contribute to the base of scientific knowledge. But the data of this manuscript cannot support the conclusions. Some technical issues have to be resolved in the manuscript.

2. This manuscript has the statistical analysis problem.

3. The revised manuscript needs to address each of the comments of the reviewers.

We look forward to receiving your revised manuscript.

Kind regards,

Tzong-Yueh Chen, Ph.D.

Academic Editor

PLOS ONE

Reviewers' comments:

Reviewer's Responses to Questions

**Comments to the Author**

1. If the authors have adequately addressed your comments raised in a previous round of review and you feel that this manuscript is now acceptable for publication, you may indicate that here to bypass the “Comments to the Author” section, enter your conflict of interest statement in the “Confidential to Editor” section, and submit your "Accept" recommendation.

Reviewer #1: (No Response)

Reviewer #2: All comments have been addressed

2. Is the manuscript technically sound, and do the data support the conclusions?

Reviewer #1: Partly

Reviewer #2: Yes

3. Has the statistical analysis been performed appropriately and rigorously? 

Reviewer #1: I Don't Know

Reviewer #2: Yes

4. Have the authors made all data underlying the findings in their manuscript fully available?

Reviewer #1: Yes

Reviewer #2: (No Response)

5. Is the manuscript presented in an intelligible fashion and written in standard English?

Reviewer #1: Yes

Reviewer #2: Yes

6. Review Comments to the Author

Reviewer #1: The revision has improved and given more clues in the potential bacterial community change. However, come technical issues have to be resolved or addressed in the context.

1. ARISA: I thank the author for his/her detailed reply on how ARISA was used in previous studies. Nonetheless, most of the studies using ARISA were back in the time NGS was not widely used. The current bacterial studies would need NGS (16S rDNA) analysis to really point out the bacterial community change. It would be more technical sound if the author can show a data to indicate that each ARISA peak really represents one bacterial species/genera in order to reach NGS resolution. (Is the binning method good enough to separate single bacteria type?)

2. Can the author explain how this is happening in my previous comment#4 regarding the highly variable bacterial community in feeds but later disappeared in the intestine?

3. Have author compared the ARISA peaks to parental fish from the early- and late-stage larva? I'm asking because part of the bacterial community may be acquired from the parents.

4. Can the author discuss the error by DNA extraction and the resulting potential error in bacterial community diversity/richness regarding to my previous comment# 6?

5. It would be good to put a rough estimate of bacterial change in phyla based on ARISA peak change (eg. from Gamma-proteobacteria to alpha-proteobacteria based on length) for NGS substitution.

6. As the author mentioned. OTU 429.1 is of my interest. Is there any possible way to know what bacteria it is? It would be powerful to show this bacteria genera/species to consolidate your conclusion in succession of bacterial community.

7. Fig S3 (b) : Please indicate color difference (What is yellow and brown group?).

8. Line 266-267 : The sentence "the dominant bacteria are likely to have a great impact on host development, as can be seen from probiotic strategy" is overstated here without evidence based data.

Reviewer #2: The manuscripte had been reviced, all commands have been addressed, and seems there are no more questions from me.

7. PLOS authors have the option to publish the peer review history of their article (what does this mean?). If published, this will include your full peer review and any attached files.

Reviewer #1: No

Reviewer #2: No

---

## [Author Response · Author response to Decision Letter 1]

26 Aug 2022

PONE-D-21-35860R1

Succession of the intestinal bacterial community in Pacific bluefin tuna (Thunnus orientalis) larvae

PLOS ONE

1. The data cannot support the conclusions. PLOS ONE is designed to communicate primary scientific research, and welcome submissions in any applied discipline that will contribute to the base of scientific knowledge. But the data of this manuscript cannot support the conclusions. Some technical issues have to be resolved in the manuscript.

> In this manuscript, we aimed to know how the pattern of intestinal bacteria change as the host fish grow. Hence, we think that the method ARISA and obtained data are enough to address it and support the conclusions.

2. This manuscript has the statistical analysis problem.

> We have already addressed statistical problems pointed out by two reviewers. Reviewer #1, who initially indicated that there was a problem, no longer indicate it, and Reviewer's Responses to Questions #3 in this time is also “no problem”.

3. The revised manuscript needs to address each of the comments of the reviewers.

> We have responded to the reviewers’ comments individually.

Reviewer #1: The revision has improved and given more clues in the potential bacterial community change. However, come technical issues have to be resolved or addressed in the context.

> We would like to thank the Reviewer for the careful review of this manuscript.

1. ARISA: I thank the author for his/her detailed reply on how ARISA was used in previous studies. Nonetheless, most of the studies using ARISA were back in the time NGS was not widely used. The current bacterial studies would need NGS (16S rDNA) analysis to really point out the bacterial community change. It would be more technical sound if the author can show a data to indicate that each ARISA peak really represents one bacterial species/genera in order to reach NGS resolution. (Is the binning method good enough to separate single bacteria type?)

> In this manuscript, we aimed to investigate how intestinal microbiota communities are formed with respect to microbes found in food and water. We know your point, however ARISA allows us for the rapid comparison with high-throughput and cost-effective, and can also yield meaningful and valuable results. Previous studies show that each ARISA peak represents one bacterial species or strain and the binning method good enough to separate single species (e.g., Brown et al. 2005, Kovacs et al. 2010). Also, several reports show ecological coherence of bacterial diversity pattern between ARISA and NGS techniques (e.g., Bienhold et al. 2011, Gobet et al. 2014, Jami et al. 2014). 

Thus, as you suggested, we added one sentence on the ecological coherence between the techniques: lines 253–255 in Revised_Manuscript “It is also reported that there is ecological coherence of bacterial diversity pattern between ARISA and deep sequencing techniques [36–38]”.

2. Can the author explain how this is happening in my previous comment#4 regarding the highly variable bacterial community in feeds but later disappeared in the intestine?

> We think this probably occurs due to the development of the host gastrointestinal tract, particularly the increased activities of digestive enzymes. Only bacteria that can tolerate the intestinal environment can survive and so the most of highly variable bacterial community in feeds cannot survive and colonize in the intestine (lines 281–294 in Revised_Manuscript).

3. Have author compared the ARISA peaks to parental fish from the early- and late-stage larva? I'm asking because part of the bacterial community may be acquired from the parents.

> No, we haven’t. The specific bacteria in the late stage larvae, 17 days after hatching, were also found on egg surface, egg shell. The egg shells were floating in the rearing water for long time after hatching. If the vertical transmission of bacteria from the parent fish occurred, the bacteria from the parents might give an influence on the early stage of larvae. Also, to our knowledge, a bacterial vertical transmission has not yet been reported as for bluefin tuna and there is only one paper on betanodaviruses in bluefin tuna (Sugaya et al., 2009). Anyway, as we have no data about the intestinal bacteria of adult bluefin tuna, this point, the vertical transmission from the parents, should be discussed in the future study. Thank you.

4. Can the author discuss the error by DNA extraction and the resulting potential error in bacterial community diversity/richness regarding to my previous comment# 6?

> This was the challenge in the past as we commented previously (e.g., Daly et al., 2012; Yuan et al., 2012; Eun et al., 2014; Dubin et al., 2016; Han et al., 2018). The results of DNA extraction are controversial even now as you know, and there may or may not be some errors. Therefore, readers interested in such paper should know this possibility, as all DNA extraction methods contain some errors.

5. It would be good to put a rough estimate of bacterial change in phyla based on ARISA peak change (eg. from Gamma-proteobacteria to alpha-proteobacteria based on length) for NGS substitution.

> Unfortunately, we cannot estimate this bacterial change. As you suggested, we are now planning a new experiment to compare the bacterial communities in not only the larvae but also the adult fish. This is because the present study showed us the important period of the formation of intestinal bacteria.

6. As the author mentioned. OTU 429.1 is of my interest. Is there any possible way to know what bacteria it is? It would be powerful to show this bacteria genera/species to consolidate your conclusion in succession of bacterial community.

> Unfortunately, we cannot know this bacterial species. However, one of the important results in this manuscript is that bacteria in feeds, controlled by the host fish, do not necessarily establish intestinal bacterial community. We think that the findings obtained in this manuscript are also valuable and appropriate for this journal.

7. Fig S3 (b) : Please indicate color difference (What is yellow and brown group?).

> We have addressed it. Thank you. “(b) PCoA constructed using only intestinal samples, the early (yellow group) and late (brown group) stages based on Adonis test (p = 0.0001).”

8. Line 266-267 : The sentence "the dominant bacteria are likely to have a great impact on host development, as can be seen from probiotic strategy" is overstated here without evidence based data.

> We have toned down this statement as follows: lines 254–255 "The dominant bacteria may have a great impact on host development, as can be seen from probiotic strategy"

---

## [Decision Letter · Decision Letter 2]

13 Sep 2022

Succession of the intestinal bacterial community in Pacific bluefin tuna (Thunnus orientalis) larvae

PONE-D-21-35860R2

Dear Dr. Eguchi,

We’re pleased to inform you that your manuscript has been judged scientifically suitable for publication and will be formally accepted for publication once it meets all outstanding technical requirements.

Kind regards,

Tzong-Yueh Chen, Ph.D.

Academic Editor

PLOS ONE

Additional Editor Comments (optional):

Reviewers' comments:

Reviewer's Responses to Questions

**Comments to the Author**

1. If the authors have adequately addressed your comments raised in a previous round of review and you feel that this manuscript is now acceptable for publication, you may indicate that here to bypass the “Comments to the Author” section, enter your conflict of interest statement in the “Confidential to Editor” section, and submit your "Accept" recommendation.

Reviewer #1: All comments have been addressed

2. Is the manuscript technically sound, and do the data support the conclusions?

Reviewer #1: Yes

3. Has the statistical analysis been performed appropriately and rigorously? 

Reviewer #1: Yes

4. Have the authors made all data underlying the findings in their manuscript fully available?

Reviewer #1: Yes

5. Is the manuscript presented in an intelligible fashion and written in standard English?

Reviewer #1: Yes

6. Review Comments to the Author

Reviewer #1: The author has addressed the limitations of ARISA methods as well as aiming for the future experiments and added references and discussion in the context. I believe this manuscript is now ready for publication.

7. PLOS authors have the option to publish the peer review history of their article (what does this mean?). If published, this will include your full peer review and any attached files.

Reviewer #1: No

---

## [Editor Report · Acceptance letter]

19 Sep 2022

PONE-D-21-35860R2 

Succession of the intestinal bacterial community in Pacific bluefin tuna (*Thunnus orientalis*) larvae 

Dear Dr. Eguchi:

I'm pleased to inform you that your manuscript has been deemed suitable for publication in PLOS ONE. Congratulations! Your manuscript is now with our production department. 

Kind regards, 

on behalf of

Prof. Tzong-Yueh Chen 

Academic Editor

PLOS ONE